# Improving Protein Interaction Prediction Using Pretrained Structure Embedding

## Abstract

The study of protein-protein interactions (PPIs) plays an important role in the discovery of protein drugs and in revealing the behavior and function of cells. So far, most PPI prediction works focus on protein sequence and PPI network structure, but ignore the structural information of protein physical binding. This results in interacting proteins are not similar necessarily, while similar proteins do not interact with each other. In this paper, we design a novel method, called PSE4PPI, which can leverage pretrained structure embedding that contain further structural and physical pairwise relationships between amino acid structure information. And this method can be transferred to new ppi predictions, such as antibody-target interactions and PPIs across different species. Experimental results on PPi predictions show that our pretrained structure embedding leads to significant improvement in PPI prediction comparing to sequence and network based methods. Furthermore, we show that embeddings pretrained based on ppi from different species can be transferred to improve the prediction for human proteins.

## 1 Introduction

Proteins are the basic functional units of human biology. However, they rarely function alone and usually do so in an interactive manner. Protein-protein interactions (PPIs) are important for studying cytoomics and discovering new putative therapeutic targets to cure diseases Szklarczyk et al. (2015). But these research processes usually require expensive and time-consuming wet experimental results to obtain PPI results. The purpose of PPI prediction is to predict there exists protein physical binding for a given pair of amino acid sequences of proteins or not.

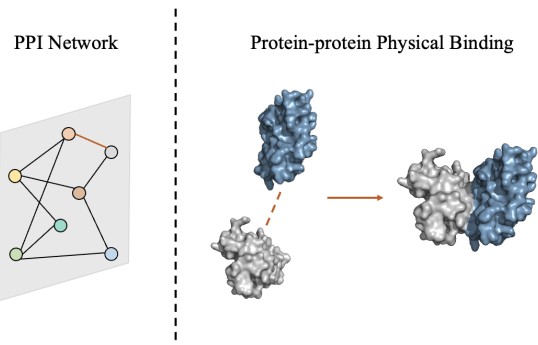

Figure 1: The illustration of PPI network and Protein-protein physical binding.

Recently, most PPI prediction works focus on protein sequence Sun et al. (2017); Hashemifar et al. (2018); Zhang et al. (2019) and PPI network structure Hamilton et al. (2017); Yang et al. (2020), which can obtain the representation of proteins by the amino acid sequence of proteins and the local neighborhood structure of PPI network, and then calculate whether there is an interaction relationship between proteins. And the PPI prediction problem is often formalized as link prediction

problem and the similarity between two proteins is calculated to predict whether the bind will formulate. But researchers report that interacting proteins are not similar necessarily, while similar proteins do not interact with each other Kovács et al. (2019). This is because the above methods ignore the influence of protein structure information on PPi prediction problems.

We need to reconfirm that the relation type of PPIs mentioned in this paper are limited to physical binding. So it is not enough to consider the similarity between the sequence level and the structure level of PPI network for the physical binding between proteins, because it requires the two proteins to interact at the 3D-structure level. Predicting the physical binding between proteins need interface contact, and then the corresponding amino acids to formulate physical binding. Therefore, we introduce protein structure information through pretrained structure embedding to improve the prediction accuracy of PPI.

In this paper, we first get our pretrained structure embeddings from pretrained protein structure model Wu et al. (2022) in PPI network, which contains the protein sequence information and the physical pairwise relationships between amino acids. Then, we use GNN-based method to predict PPI networks, in which we consider proteins as nodes, interactions as edge, and pretrained structure embeddings as features of nodes. In addition, we also use data from different species in the StringDB as pretraining data for human PPI prediction, and explore whether PPI prediction is transferable across different species.

Our contribution include (1) To our best knowledge, we are first to use pretrained protein structure embedding as PPI network feature to solve PPI prediction problems; (2) We try to use StringDB Mering et al. (2003) to introduce more PPI data of the same or different species to improve the performance of the prediction model. (3) We used different species PPIs as pretraining data, and find that different species PPI data are significantly helpful for PPI prediction of human; (4) We achieved the SOTA performance among GNN-based PPI prediction methods.

## 2 RELATED WORK

### 2.1 SEQUENCED-BASED PPI METHOD

The sequential input-based model can use high and complex input features to improve the prediction ability of the model for PPI. Sun et al. (2017) first applied stacked autoencoder (SAE) to obtain sequence-based input features in sequence-based PPI prediction problem. Du et al. (2017) proposed a deep learning-based PPI model (DeepPPI), which can obtain high-level features from the general description of proteins. Hashemifar et al. (2018) proposes a convolution-based model that inputs the sequences of a bunch of proteins to predict whether they will interact; Gonzalez-Lopez et al. (2018) proposed a method that can use recurrent neural networks to process raw protein sequences and predict protein interactions without relying on feature engineering. Jha et al. (2021) proposed a deep multimodal framework to predict protein interactions using structural and sequence features of proteins and due to existing work.

### 2.2 GNN-BASED METHOD

Graph neural network plays a very important role in solving protein interaction problems through PPI network. Yang et al. (2020) studied PPI prediction based on both sequence information and graph structure, showing superiority over existing sequence-based methods. Baranwal et al. (2022) proposed a graph attention network for structure based predictions of PPIs, named Struct2Graph, to describe a PPI analysis. Colonnese et al. (2021) adopted a Graph Signal Processing based approach (GRABP) modeling PPI network with a suitably designed GSP (Graph Signal Processing) based Markovian model to represent the connectivity properties of the nodes. Kabir & Shehu (2022) introduced a novel deep learning framework for semi-supervised learning over multi-relational graph, which learned affect of the different relations over output. Jha et al. (2022) combined molecular protein graph neural network (GNN) and language model (LM), by generating per-residue embedding from the sequence information as the node's feature of the protein graph, to predict the interaction between proteins

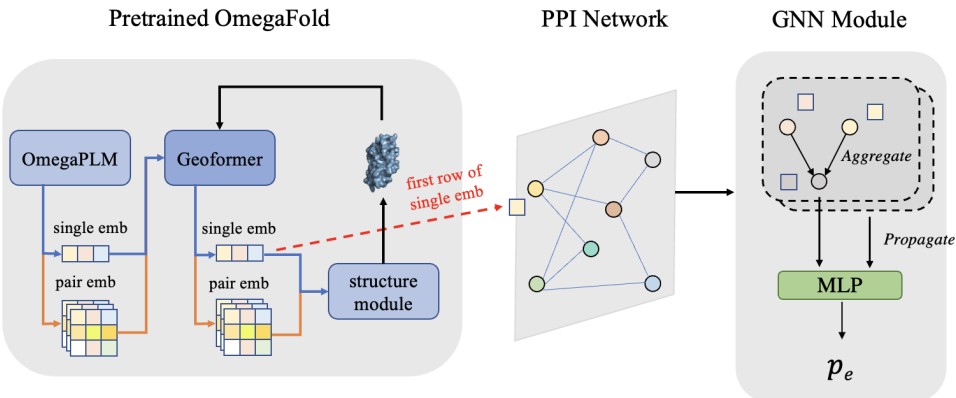

Figure 2: The overview of PSE4PPI, which can leverage pretrained structure embedding from the first row of single embedding in Geoformer output that contain further structural and physical pair-wise relationships. Then, we regard the pretrained structure embeddings as the features of the nodes in the PPI network, and then use the GNN-based method to obtain the representation of the nodes to predict PPI.

## 2.3 PRETRAINED LANGUAGE MODEL

Pre-trained models can be used to extract features from large biological sequences. Some studies Playe & Stoven (2020); Zhuang et al. (2019)use CNNS to pretrain biological sequences in transfer learning. Zhuang et al. (2019) proposed a simple CNN model based on DNA sequences to predict EPI. Another neural network that can be used to process sequence information, LSTM, has been proposed to represent biological sequences as pretrained models Strodthoff et al. (2020); Bepler & Berger (2019); Villegas-Morcillo et al. (2021).

In recent years, many Transformer based models Rao et al. (2020); Rives et al. (2021); Rao et al. (2021); Ruffolo et al. (2021) have been shown to serve as pre-trained models for embedded representation of biological sequences, especially after Bert Devlin et al. (2018) was proposed. Lin et al. (2022) used relative position encoding to capture the pair-wise information between amino acids in the sequence. Wu et al. (2022) used OmegaPLM and self-attention mechanism to capture co-evolution information.

## 3 METHODOLOGY

We begin to introduce our method of using pretrained structure embedding for improving protein interaction prediciton (PSE4PPI), as shown in Figure 2. PSE4PPI uses structure embeddings extracted from pretrained protein structure prediction models, OmegaFold Wu et al. (2022), as node feature in PPI network to improve the accuracy of GNN in predicting protein interactions. We also use StringDB to provide more PPI data for multiple species to pretrain to acquire the general knowledge across different species.

In the following, we will first present overview of PSE4PPI, and then introduce how to extract structural embeddings using pretrained OmegaFold. Then we introduce how to use protein structures as edge features in PPI networks for PPI prediction. Finally, we describe how PPI data across species can be used to learn general knowledge in PPI prediction.

## 3.1 OVERVIEW OF PSE4PPI

To obtain the pretrained protein structure embeddings, protein sequences can be input into the pretrained protein language model (OmegaPLM) and obtain the residue-level node embeddings and residue-residue pairwise embeddings. And then through multiple layers of Geoformer, a new geometric inspiration transformer Vaswani et al. (2017), to further extract structural and physical pair-

wise relationships between amino acid structure information. Finally, a structure prediction module can predict the 3D spatial coordinates of the protein from the final output of Geoformer. The final predicted protein structure and embeddings will be recycled into a new round of prediction to calculate a more refined protein structure. We extract the first line of the residue embedding output by Geoformer in the last cycle as the pretrained structure Embedding of this protein sequence, because the embedding we take out is the embedding used in protein structure prediction.

After acquire the pretrained structure embeddings of proteins, PPIs can be constructed as a network $\mathcal{G} = \{\mathcal{V}, \mathcal{E}\}$, namely a PPI network, in which proteins are regarded as nodes, protein interactions as edges, and the pretrained structure embeddings of proteins are the features of nodes. Then we can use graph neural networks as encoders, such as GraphSAGE and GAT , to learn the node representation in the PPI network. Finally, the fully connected network or CNN was used as decoder to predict whether physical binding would be formed between the two proteins.

The model also supports training in a larger PPI network and then transfering to a new PPI network for PPI prediction. For example, we pretrain the GNN-based model in the PPI network of human proteins and then fine-tune it in the antibody-target interaction network or the PPI network of other species. And the improvement in this setting is obvious, which alleviates the problem caused by insufficient PPI network data.

## 3.2 PRETRAINED STRUCTURE EMBEDDING

Omegafold contains three modules that OmegaPLM, Geoformer and Structure Module. OmegaPLM generates a residue-level representation with a single sequence as input by default. At the same time, the weight of the multi-head attention mechanism in the pretrained language model contains the relationship between residues in the protein sequence, which can naturally be regarded as the initial residue-level pair representation. Formally, given a sequence of a protein, we can get the residue embedding and pair embedding:

$$
\begin{aligned}
\boldsymbol{z}_{plm}, \boldsymbol{w}_{plm} &= OmegaPLM(\boldsymbol{s}), \\
\boldsymbol{z}_{plm} &\in \mathbb{R}^{N_{MSA} \times N_s \times d^{plm}}, \\
\boldsymbol{w}_{plm} &\in \mathbb{R}^{d^{stack} \times N_s \times N_s},
\end{aligned}
\tag{1}
$$

where $\boldsymbol{z}_{plm}$ is the single representation, $\boldsymbol{w}_{plm}$ is the pair representation, $\boldsymbol{s} = \{s_1, s_2, ..., s_{N_s}\}$ is the sequence of the protein, $N_{MSA}$ is default number of MSA sequences in AlphaFold2, $d^{stack}$ is the self-attention layers of OmegaPLM, $d^{plm}$ is the node dimension of pretrained language model.

Before input the single embedding and pair embedding into Geoformer, we need to adjust the dimensions of $\boldsymbol{z}_{plm}, \boldsymbol{w}_{plm}$ by linear functions:

$$
\begin{aligned}
\boldsymbol{z}_{plm}' &= Linear(\boldsymbol{z}_{plm}), \\
\boldsymbol{w}_{plm}' &= Linear(\boldsymbol{w}_{plm}), \\
\boldsymbol{z}_{plm}' &\in \mathbb{R}^{N_{MSA} \times N_s \times d_n^{Geo}}, \\
\boldsymbol{w}'_{plm} &\in \mathbb{R}^{d_e^{Geo} \times N_s \times N_s},
\end{aligned}
\tag{2}
$$

where $d_n^{Geo}, d_e^{Geo}$ are the dimension of node embedding and edge embedding in Geoformer. After this, OmegaFold uses Geoformer to further calculate the sequence and structural information of the amino acids in the protein,

$$
\begin{aligned}
\boldsymbol{z}_{geo}, \boldsymbol{w}_{geo} &= Geoformer(\boldsymbol{z}_{plm}', \boldsymbol{w}_{plm}'), \\
\boldsymbol{z}_{geo} &\in \mathbb{R}^{N_{MSA} \times N_s \times d_n^{geo}}, \\
\boldsymbol{w}_{geo} &\in \mathbb{R}^{d_e^{geo} \times N_s \times N_s},
\end{aligned}
\tag{3}
$$

where $\boldsymbol{z}_{geo}, \boldsymbol{w}_{geo}$ are single embedding and pair embedding of Geoformer output. Finally, the structure module will use the first row of single embedding $\boldsymbol{z} = \boldsymbol{z}_{geo}[0]$ and pair embedding to predict the 3D structure of protein. In the final cycle of recycling procedure, we use the $\boldsymbol{z}$ as the protein structure embedding.

### 3.3 PPI PREDICTION

After the protein pretrained structure embedding is obtained, the representation of nodes in PPI network $\mathcal{G} = \{\mathcal{V}, \mathcal{E}\}$ can be calculated by graph neural network(GNNs). It should be noted that the $\boldsymbol{z} \in \mathbb{R}^{N_s \times d_n^{geo}}$ and the sequence length $N_s$ of nodes will change due to different proteins, so the features of nodes should be pooling before the neighbor aggregation. We will conduct mean-pooling along the dimension of sequence length, and then we will get the features $\boldsymbol{z}^g$ of nodes that can participate in neighbor aggregation.

Most GNNs basically follow similar operations, $i.e.$, node features aggregation and transform the message along the network topology for a certain layers. Formally, the $k - th$ layer representation of node $v \in \mathcal{V}$ can be represented as:

$$
\begin{aligned}
\boldsymbol{h}^k &= \mathbf{PROPAGATE}(\mathbf{z^g}; \mathcal{G}; k) \\
&= \left\langle \mathbf{Trans}\left(\mathbf{Agg}\{\mathcal{G}; \mathbf{h}^{k-1}\}\right)\right\rangle,
\end{aligned}
\tag{4}
$$

with $\boldsymbol{h}^0 = \boldsymbol{z}^g$ and $\boldsymbol{h}$ is the output representation after the $k - th$ layer graph neural network. $\mathbf{Agg}\{\mathcal{G}; \boldsymbol{h}^{k-1}\}$ means aggregating the $(k - 1) - th$ layer result $\boldsymbol{h}^{k-1}$ along the network $\mathcal{G}$ for the $k - th$ aggregate operation, $i.e.$, mean-pooling or using attention mechanism, and $\mathbf{Trans}(\cdot)$ is layer-wise feature transformation operation including weight matrix and non-linear activation function, $i.e.$, sigmoid or ReLU.

When it comes to predicting whether physical binding between two proteins will establish, the problem is transformed into link prediction problem between two nodes $v_1, v_2$ in a PPI network. To solve this problem, we concatenate the representations $\boldsymbol{h}_1^k, \boldsymbol{h}_2^k$ of nodes $v_1, v_2$ together as the input of fully connected network, and the output was to predict the possibility of establishing physical binding between two nodes,

$$
p_e = \sigma(\mathbf{MLP}\left(\boldsymbol{h}_1^k \| \boldsymbol{h}_2^k\right)),
\tag{5}
$$

where $p_e$ is the probability that $v_1, v_2$ can establish a link, $\sigma$ is the $sigmoid$ function, $\mathbf{MLP}$ can be two-layer MLP, and $\|$ is the concatenation operator.

### 3.4 OBJECTIVE FUNCTION

Because there are only physical binding relationships between proteins in the network, we need to randomly sample negative examples without physical binding relationships across the network. We minimize the loss for each PPI network edge $e \in \mathcal{E}$ to train the model:

$$
\mathcal{L} = -y_e \log p_e - (1 - y_e) \log(1 - p_e),
\tag{6}
$$

where $y_e = 1$ if $e$ exists in edge set $\mathcal{E}$ and $y_e = 0$ otherwise.

## 4 EXPERIMENT

We conduct experiments on two different datasets of PPI networks, including antibody-target interaction prediction and membrane protein interaction prediction of human. We also test the performance of the model pretrained on human PPI networks and prediction on other PPI network of different spicies. We compare the proposed method PSE4PPI with two types of baseline methods on the PPI prediction task, and present experimental results to answer three research questions: (RQ1) How does PSE4PPI performs compared with SOTA methods? (RQ2) Whether the pretrained structure embedding helps PPI prediction? (RQ3) How useful is pretraining on PPI data across species for PPI prediction?

Table 1: Statistics of datasets.

| Dataset | ATI | M2H-PPI | H-PPI | AM-PPI |
|---|---|---|---|---|
| **Unique nodes** | 2 | 624 | 11403 | 545 |
| **Protein Interactions** | 4133 | 39041 | 156066 | 14664 |

## 4.1 EXPERIMENTAL SETUP

**Dataset**   In order to test the prediction effect of PSE4PPI on PPI, Antibody-target Interaction Prediction (ATI) dataset, Membrane Protein Interaction dataset of Human (M2H-PPI) and human PPI network (H-PPI) are selected. In order to test how helpful the pretraining of PPI data across species is for PPI prediction, we choose the H-PPI as pretraining dataset and fintune on (ATI) dataset, M2H-PPI dataset and Actinosynnema mirum PPI dataset (AM-PPI). Among them, M2H-PPI, H-PPI, AM-PPI dataset are from StringDB. The statistics of the dataset are summarized in Table 1

**Baselines**   In order to verify that our pretrained structural embeddings are effective, we adopt two two types of methods to deal with them:  **(1) Sequence-based method**  : Resnet ; **(2) GNN-based method** : GraphSAGE, GAT.

**Implementations**   In order to measure the performance of the model in the PPI network, we employ three metrics to fully evaluate our model and baseline methods: accuracy (ACC), AUC and Micro-F1. We employ Adam optimizer for three types of methods' training, and the maximum number of epoch is set to 1000. (1) For the setting of the ResNet, we use 6-layer ResNet for prediction the pretrained structure embedding. For all four datasets, we set batch size as 8192 and embedding dimension as 256. We set the learning rates for ResNet $lr_{Res} = 0.005$, and dropout is 0.05. (2) For the setting of GraphSAGE and GAT, we use 2-layer GNN as encoder and 2-layer MLP as decoder.We set the batch size as 512, hidden size is 512, learning rate $lr_{GNN} = 0.001$, dropout is 0.1, the number of aggregated neighbors is 10. We conduct experiments on a Linux server with a single GPU (GeForce RTX) and CPU (AMD) with PyTorch 1.4.0.

Table 2: Experimental results of PPI prediction on four datasets.  Larger scores indicate better performances.

|  | ATI | | | M2H-PPI | | | H-PPI | | | AM-PPI | | |
|---|---|---|---|---|---|---|---|---|---|---|---|---|
|  | *ACC* | *AUC* | *F1* | *ACC* | *AUC* | *F1* | *ACC* | *AUC* | *F1* | *ACC* | *AUC* | *F1* |
| **GraphSAGE** | 84.64 | 86.21 | 85.37 | 81.32 | 88.07 | 80.89 | 83.65 | 90.12 | 82.82 | 81.81 | 87.32 | 70.74 |
| **GAT** | 84.87 | 86.94 | 85.82 | 81.97 | 88.84 | 80.49 | 83.12 | 90.27 | 82.16 | 82.79 | 88.04 | 71.69 |
| **PSE4PPI-SAGE** | 88.21 | 89.15 | 88.64 | 85.63 | 92.92 | 84.32 | 86.65 | 93.12 | 85.82 | 85.21 | 91.14 | 74.12 |
| **PSE4PPI-GAT** | 89.21 | 90.60 | 85.82 | 85.82 | 92.98 | 84.39 | 86.81 | 93.32 | 86.91 | 86.14 | 92.02 | 75.13 |

## 4.2 MAIN RESULTS ON PPI PREDICTION (RQ1)

In this subsection, we compare our main experiment results on four datasets using pretrained structure embeddings with randomly initialized features. The results in Table 2 show that the pretrained structure embedding can effectively help the GNN-based method to make PPI prediction in PPI network. As mentioned above, it is biased to predict PP using the method based on similarity measure in PPI network because it only considered the topology of PPI network and ignored the structure information about protein itself. We use the trained protein structure prediction model to extract protein structure embeddings and treat them as the features of the nodes in the PPI network. In this way, when we conduct neighborhood aggregation along the relationship in the PPI network through GNN, we not only capture the topological information in the PPI network, but also use the

sequence and structure information of the protein itself. Therefore, we can use pretrained structural embeddings as node features to effectively help GNN-based methods improve PPI prediction ability.

## 4.3 ANALYSIS OF PRETRAINED STRUCTURE EMBEDDING (RQ2)

In this subsection, we compare the difference in performance resulting from extracting structural embeddings using OmegaPLM and OmegaPLM + GeoFormer on two datasets. Table 3 shows that the pretrained structure embedding extracted from OmegaPLM + Geoformer provides more obvious help on CNN-based method and GNN-based method compared with that extracted only by OmegaPLM. It is obviously that OmegaPLM learns coevolution information of protein sequences by stacking self-attention layers, and Geoformer further extracts the relationship between protein structure and physical interactions. Therefore, OmegaPLM+Geoformer can further improve the PPI prediction ability of the model.

We believe that the pretrained structure embedding obtained by Geoformer could be more helpful to the final PPI prediction result than the current experiment. We speculate that it may be necessary to add a decoder such as convolutional network before using the structure embedding as the node feature, which will be our further study on this issue.

Table 3: Experimental results of the influence of OmegaPLM and Geoformer on structure embedding. Larger scores indicate better performances.

| | OmegaPLM | Geoformer | ATI | | | H-PPI | | |
|---|---|---|---|---|---|---|---|---|
| | | | *ACC* | *AUC* | *F1* | *ACC* | *AUC* | *F1* |
| **PLM-ResNet** | ✓ | | 88.37 | 89.32 | 88.59 | 83.65 | 91.47 | 85.70 |
| **PLM-GraphSAGE** | ✓ | | 88.69 | 88.94 | 88.31 | 83.82 | 91.71 | 85.89 |
| **PLM-GAT** | ✓ | | 88.42 | 89.94 | 88.72 | 84.79 | 92.13 | 86.97 |
| **PSE4PPI-ResNet** | ✓ | ✓ | 89.17 | 90.52 | 89.91 | 86.23 | 93.18 | 85.52 |
| **PSE4PPI-SAGE** | ✓ | ✓ | 89.92 | 91.51 | 90.67 | 86.65 | 93.12 | 85.82 |
| **PSE4PPI-GAT** | ✓ | ✓ | 89.98 | 91.73 | 90.97 | 86.81 | 93.32 | 86.91 |

## 4.4 PPI PREDICTION RESULTS OF PRETRAINING USING DIFFERENT SPECIES (RQ3)

In this subsection, we want to test whether our model can be pretrained on PPI data of different species to improve the prediction performance on another PPI dataset. As shown in Table 4, we used H-PPI data to pretrain the PSE4PPI model, and then finetune and test it on ATI and AM-PPI data. The results on ATI data can reflect the model pretrained on H-PPI data, which can help the prediction on ATI data to a certain extent. But our attempts on the AM-PPI dataset did not get the results we wanted.

We speculate that the task of pretraining is simple and single compared with the general pre-training strategy, which may not be able to capture the general knowledge of PPI that can cross species well. Moreover, pretraining on PPI data of one species with the hope of migrating to another species will also be relevant for the selected species, and these questions will be carried out step by step as our research progresses.

## 5 CONCLUSION

In this paper, we have presented a novel method, named PSE4PPI, which can which can leverage pretrained structure embedding that contain further structural and physical pairwise relationships between amino acid structure information. Our experiments demonstrate that our use of pre-trained structural embeddings to predict PPI is obvious compared to using only the topological information

Table 4: About the effect of pre-training on H-PPI data on the new PPI task. Larger scores indicate better performances.

| | ATI | | | AM-PPI | | |
|---|---|---|---|---|---|---|
| | *ACC* | *AUC* | *F1* | *ACC* | *AUC* | *F1* |
| **PSE4PPI-ResNet** | 89.17 | 90.52 | 89.91 | 85.49 | 90.91 | 74.05 |
| **PSE4PPI-GraphSAGE** | 89.92 | 91.51 | 90.67 | 85.21 | 91.14 | 74.12 |
| **PSE4PPI-GAT** | 89.98 | 91.73 | 90.97 | 86.14 | 92.02 | 75.13 |
| **pretrained-PSE4PPI-ResNet** | 90.32 | 91.74 | 91.16 | - | - | - |
| **pretrained-PSE4PPI-SAGE** | 90.64 | 92.31 | 91.02 | - | - | - |
| **pretrained-PSE4PPI-GAT** | 90.72 | 92.63 | 91.36 | - | - | - |

of PPI networks. Moreover, the model can be finetuned on PPI data of another species using a model pretrained on the PPI network of one species.

**Limitations and Broader Impact**    The limitation of PSE4PPI is that it is rough to use the pretrained structure embedding directly as the features of the nodes in the PPI network, because it is difficult to decode the extracted structure embeddings in the protein structure prediction model with several layers of GNN network and MLP. Therefore, we speculate that the pretrained structure embedding can be decoded through a convolutional network before being fed into the GNN-based method. In addition, our model does not perform very well when transferring to new protein prediction problems, a large part of the reason is related to our training task and pretraining data. These limitations will be the direction of our further research. For the first time, we tried to take out the structure embedded in the protein structure prediction model to assist PPI prediction. And it remains to be seen whether a large of PPIs can in turn help protein structure prediction.

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
