# OpenReview forum: "Improving Protein Interaction Prediction using Pretrained Structure Embedding"
_ICLR.cc/2023/Conference — Submitted to ICLR 2023_

### Official Review · Reviewer_3nJt · 2022-10-20

**Confidence:** 4
**Clarity, Quality, Novelty And Reproducibility:** The idea of using OmegaFold embedding…
**Correctness:** 1
**Technical Novelty And Significance:** 2
**Empirical Novelty And Significance:** 1
**Recommendation:** 3

**Strength And Weaknesses:**

+ Multiple datasets considered
- Poor baseline choices
- Unclear model design - advantage of both structural embeddings and GNN aggregation is not supported by results


Questions:
How does the model compare to an AlphaFold-Multimer based PPI prediction approach?
How does the model compare to a baseline that just uses OmegaFold embeddings pairwise without any GNN?
How do OmegaFold embeddings compare to just directly using OmegaPLM embeddings? This is a relevant comparison as the paper tries to prove that using structural embeddings is advantageous.

The paper exhaustively discusses OmegaFold, which is distracting given that this is previous work and is just used as a pretrained model without any modifications.

**Summary Of The Paper:**

The paper proposes to use pretrained protein representations for predicting whether two given proteins interact (protein-protein interaction, PPI). Specifically, it proposes to use pretrained embeddings from the encoder part of the OmegaFold protein structure prediction model. For the trained prediction network, a GNN operating on PPI networks is proposed, where the OmegaFold embeddings are the node features and the task is to predict whether an edge between two proteins in the network exists. As known PPI networks by definition only contain true edges, negative sampling is applied during training and evaluation to augment the network with negative edges.





**Summary Of The Review:**

While the basic idea of using protein structural embeddings for PPI is well motivated, the paper does a poor job at proving that the proposed model indeed works. First of all, the baseline for comparison is arbitrarily weak and certainly not the previous state of the art. PPI prediction is a hard task, and a simple ResNet trained directly on protein sequences cannot be expected to perform particularly well. It is therefore impossible to tell from the presented results how the model compares to other PPI approaches. It is especially worrying that any comparison to AlphaFold-Multimer is missing, both in the results and in the discussion. A starting point could be Bryant 2022 (https://doi.org/10.1038/s41467-022-28865-w)

The second issue is the GNN layer of the model. The paper presents itself as a method to predict pairwise interactions between proteins. However, instead of taking in two proteins at a time as input, the model considers a network of proteins, each represented by its structural embedding. The network is then optimized to predict whether the edges between proteins in the network are true or false, which requires random negative sampling during training. First of all, it is not clear how exactly this negative sampling was done. Additionally, it is unclear why the GNN layer and the network are needed at all, when the task is to predict the interaction between two proteins only. If the problem is cast in this way, it is necessary to show that this improves performance over a model just operating on pairwise inputs.

---

### Official Review · Reviewer_AAa4 · 2022-10-22

**Confidence:** 4
**Correctness:** 2
**Technical Novelty And Significance:** 2
**Empirical Novelty And Significance:** 2
**Recommendation:** 3

**Clarity, Quality, Novelty And Reproducibility:**

Clarity: The presentation and the writing of the paper could be largely improved. For example, what does the AUC actually measure in the protein-protein interaction context? How is the binary classification problem set up here?  More transition between the paragraphs would also help. The tables are also not in a most easily digestable format and are without confidence intervals. Changing some tables to figures might help present the data better.

Quality: On the empirical evaluation, while I'm not an expert on PPI prediction, it seems like the authors only compared to GraphSAGE and GAT. Is there any supporting evidence for them being the SOTA? I'm aware of other works such as dMaSIF for PPI prediction. The proposed method seems to be only marginally better than GAT/GraphSAGE.

Originality: Limited novelty in the technical contributions and in the application.

**Strength And Weaknesses:**

Strengths:
+ Using protein structural embedding for PPI prediction is an important direction.

Weaknesses:
+ I find this paper to be not very clear. For example, what does the AUC actually measure in the protein-protein interaction context? How is the binary classification problem set up here?
+ The empirical results only show very marginal improvements, and do not have confidence intervals.
+ While I'm not an expert on PPI prediction, it seems like the authors only compared to GraphSAGE and GAT. Is there any supporting evidence for them being the SOTA? I'm aware of other works such as dMaSIF for PPI prediction.

**Summary Of The Paper:**

This paper uses protein structure embedding as as features to predict protein-protein interaction.

**Summary Of The Review:**

PPI prediction is an important problem. However, the technical and empirical contributions in this paper are limited.

---

### Official Review · Reviewer_p7hw · 2022-10-23

**Confidence:** 4
**Clarity, Quality, Novelty And Reproducibility:** Low novelty.
**Correctness:** 4
**Technical Novelty And Significance:** 1
**Empirical Novelty And Significance:** 1
**Recommendation:** 3

**Strength And Weaknesses:**

Good
the approach is easy to combine/add to previous GNN models  and seems to provide a consistent improvement of 5%

Bad
The novelty is too limited: take a method A that provides a protein encoding (e.g. a method that is structure aware)  and use the encoding to provide additional attributes to describe the protein in a PPI GNN method.
The author take encoding methods known in literature and combine them with no significative modification of the modelling architecture.

**Summary Of The Paper:**

The authors propose to improve the task of protein-protein interaction prediction by providing a GNN approach with additional features on the proteins based on their structures and sequences.

**Summary Of The Review:**

Although simple and effective, the approach is not particularly novel nor can be used to spark new ideas in related problems.

---

### Official Review · Reviewer_23GN · 2022-10-24

**Confidence:** 4
**Correctness:** 3
**Technical Novelty And Significance:** 2
**Empirical Novelty And Significance:** 2
**Recommendation:** 3

**Clarity, Quality, Novelty And Reproducibility:**

Clarity: Mostly satisfying. Some minor issues have been listed above.

Quality: Limited contribution.

Novelty: Somehow limited. I was expecting proposing a novel protein structure pre-training method and then validate it on the PPI prediction task, rather than directly adopting OmegaFold’s embeddings and feeding into standard graph neural networks for prediction.

Reproducibility: Good. Detailed hyper-parameter settings are described.

**Strength And Weaknesses:**

Pros:
1. The motivation is well founded, since unlike link prediction in social networks, similar proteins may not interact and interacting proteins may not be similar. Thus, explicit usage of structural embeddings should be critical in accurate protein-protein interaction prediction.
2. Additional experiments are conducted to verify the transferability of PPI prediction models across different species.

Cons:
1. Limited contribution and novelty. The pre-trained structural embedding is directly adopted from OmegaFold, and it seems that the main contribution in the methodology aspect is using graph neural networks with structural embeddings as inputs for PPI prediction.
2. Since initial structural embeddings may have different length, authors adopt mean pooling along the sequence dimension to obtain fixed-length representation of each protein. Such protein-level embedding may fail to capture local structures actually participating the interaction with other proteins.
3. The results of “PSE4PPI-SAGE” and “PSE4PPI-GAT” on ATI and H-PPI datasets are not consistent in Table 2 and 3. Please clarify.

Some minor issues:
1. Table 1. Why the number of unique nodes in the ATI dataset is only 2?
2. Table 2, 3, and 4. It would be better to use bold numbers to highlight the best results in each column.
3. Based on my understanding, by default, OmegaFold does not take an actual MSA as inputs. Instead, it generates a pseudo MSA by randomly masking a certain ratio of amino-acid residues in the original sequence multiple times. Please clarify which type of MSA is used in the proposed method.
4. Additional proofreading may be needed to correct typos and grammatical mistakes in the current manuscript.

**Summary Of The Paper:**

In this paper, authors propose to adopt pre-trained structure prediction models to extract meaningful structural embeddings for protein-protein interaction (PPI) prediction. Such structural embeddings are fed into a graph neural networks to formulate the possible interaction between different proteins. Two datasets of PPI networks (antibody-target interaction and membrane protein interaction) are used to validate the effectiveness of the proposed method.

**Summary Of The Review:**

The overall motivation is reasonable and convincing, as similar proteins do not necessarily interact with each other, and additional structure information should be considered. However, the proposed method is of limited novelty and contribution, which may be less qualified to be accepted by ICLR.

---

### Decision · Program_Chairs · 2023-01-20

**Decision:**

Reject

**Justification For Why Not Higher Score:**

All reviewers and AC agree that this paper is of very limited novelty. In addition the authors did not provide any response to the reviewers comments.

**Justification For Why Not Lower Score:**

N/A

**Metareview: Summary, Strengths And Weaknesses:**

The paper proposes to solve protein-protein interaction (PPI) tasks by using embeddings produced by protein structure pertaining approaches as features.  Experiments confirm that the addition of such features is beneficial.

Strengths: The paper considers an important problem, that of PPI prediction.  The use of protein structure embeddings for PPI prediction is a sound direction that is well motivated. The presentation is clear and the paper is easy to follow.

Weakness: all reviewers and the AC agree that the novelty is too limited to warrant publication. Unfortunately the authors did not provide any response the the reviewers comments.

Important prior work is missing that employ pretrained models for PPI prediction:
Hu, Xiaotian, et al. "Deep learning frameworks for protein-protein interaction prediction." Computational and Structural Biotechnology Journal (2022).